# Degradation of DDT by a Novel Bacterium, *Arthrobacter globiformis* DC-1: Efficacy, Mechanism and Comparative Advantage

Xiaoxu Wang [1], Belay Tafa Oba [2] , Hui Wang [1,*], Qing Luo [1,3,4,5,*], Jiaxin Liu [6], Lanxin Tang [6], Miao Yang [6], Hao Wu [1] and Lina Sun [1]

1. Key Laboratory of Eco-Restoration of Regional Contaminated Environment, Ministry of Education, Shenyang University, Shenyang 110044, China; 13840554151@163.com (X.W.)
2. Department of Chemistry, College of Natural and Computational Sciences, Arba Minch University, Arba Minch 21, Ethiopia
3. Northeast Geological S&T Innovation Center of China Geological Survey, Shenyang 110034, China
4. Key Laboratory of Black Soil Evolution and Ecological Effect, Ministry of Natural Resources, Shenyang 110034, China
5. Liaoning Academy of Agricultural Sciences, Shenyang 110161, China
6. Sichuan Provincial Engineering Laboratory of Monitoring and Control for Soil Erosion in Dry Valleys, China West Normal University, Nanchong 637009, China
* Correspondence: huiwang425@126.com (H.W.); luoqingyt@126.com (Q.L.)

**Abstract:** A novel bacterium, *Arthrobacter globiformis* DC-1, capable of degrading DDT as its sole carbon and energy source, was isolated from DDT-contaminated agricultural soil. The bacterium can degrade up to 76.3% of the DDT at a concentration of 10 mg/L in the mineral salt medium (MSM) within 1 day of incubation. The effects of various environmental conditions, such as the concentration of DDT, temperature, pH and additional carbon sources, on its growth and biodegrading capacity of DDT were investigated in the MSM. The *A. globiformis* DC-1 strain could efficiently grow and degrade DDT at a wide range of concentrations, with the maximum growth and degradation rate at 10 mg/LDDT, followed by inhibitory effects at higher concentrations (20 and 30 mg/LDDT). Mesophilic temperatures (25–30 °C) and a pH of 7–7.5 were the most suitable conditions for the growth and biodegradation. The presence of carbon sources significantly increased the growth of the DC-1 strain; however, degradation was inhibited in the present of glucose, sucrose and fructose, and peptone was determined to be the most appropriate carbon source for *A. globiformis* DC-1. The optimal DDT degradation (84.2%) was observed at 10 mg/LDDT, peptone as carbon source in pH 7.5 at 30 °C with 1 day of incubation. This strain could also degrade DDE, DDD and DDT simultaneously as the sole carbon and energy source, with degradation rates reaching 70.61%, 64.43% and 60.24% in 10 days, respectively. The biodegradation pathway by *A. globiformis* DC-1 revealed that DDT was converted to DDD and DDE via dechlorination and dehydrochlorination, respectively; subsequently, both DDD and DDE transformed to DDMU through further dechlorination, and finally, after ring opening, DDMU was mineralized to carbon dioxide. No intermediate metabolites accumulation was observed during the GC/MS analysis, demonstrating that the *A. globiformis* DC-1 strain can be used for the bioremediation of DDT residues in the environment.

**Keywords:** DDT; biodegradation; *Arthrobacter globiformis*; bioremediation

## 1. Introduction

Over the last few decades, DDT (1,1,1-trichloro-2,2-bis(4-chlorophenyl)ethane) has become the most well-known and widely used insecticide in the world for preventing the spread of agricultural pests and mosquito-borne diseases, such as typhus and malaria [1,2]. DDT residues are strongly persistent and recalcitrant in the environment,

mainly in the form of DDT isomers p,p'-DDT and 1,1,1-trichloro-2-(*p*-chlorophenyl)-2-(*o*-chlorophenyl) ethane (o,p'-DDT), as well as the DDT primary metabolites 1,1-dichloro-2,2-bis(4-chlorophenyl)ethylene (p,p'-DDE) and 1,1-dichloro-2,2-bis(4-chlorophenyl) ethane (p,p'-DDD) [3]. DDT residues are classified as priority-persistent organic pollutants by the US Environmental Protection Agency (EPA) due to their toxicity, hydrophobicity and bioaccumulation [4,5]. Although DDT has been banned in most developed nations since the 1970s, and in China since 1983 [6], it is still used in some tropical developing countries as part of public health programs [7]. Furthermore, DDT has been freshly introduced into the environment from dicofol and antifouling pesticides [8]. DDT residues, which have a long half-life of 4 to 35 years, are still abundant in soil and water in many countries all over the world, especially in China [9,10]. DDT residues bioconcentrate in the adipose tissues of organisms and accumulate in food sources due to their highly lipophilic pollutants and endocrine-disrupting chemicals [11], leading to a variety of negative consequences, such as central nervous system disruption and reproductive disorders [12,13]. Given the serious threat that DDT residues pose to the environment and human health, it is critical to investigate effective remediation technology for the decontamination of DDT residues [14].

The removal of DDT from the environment has so far been studied using both physicochemical and biological remediation methods. Physical remediation, such as thermal treatment, is cost expensive, and chemical remediation, such as advanced oxidation, can produce secondary pollution [15]. Although physicochemical remediation is more rapid than biological remediation, it still requires more manpower and material resources [16,17]. Biological remediation, which utilizes microorganisms to biotransform organochlorine compounds, is an eco-friendly and cost-effective technology [18]. To date, a few DDT-degrading strains have been documented to enhance the biodegradation process under laboratory conditions [19]. For instance, Wang et al. [20] showed the capacity of the aerobic bacterium *Streptomyces* sp. D3 to degrade 77% of 20 mg/LDDT after 7 days of incubation. Neerja et al. [21] reported that *Serratia marcescens* NCIM 2919 can degrade over 42% of initial 50 mg/LDDT and other DDT metabolites, such as DDD, DDE, DDMU (1-chloro-2,2-bis(4-chlorophenyl) ethylene) and 4-chlorobenzoate (4-CBA), after a 10 day incubation period. Ito et al. [22] observed that *Streptomyces* sp. 885 removed 84.5% of 5 mg/LDDTover 14 days in the co-presence of additional carbon source. Purnomo et al. [23] found that the DDT biodegradation rate reached roughly 31% and 42% under aqueous culture conditions throughout 7 days of incubation with the bacterium strain *Ralstoniapickettii* and the brown-rot fungus *Fomitopsispinicola.* Despite the fact that these strains can effectively degrade DDT, they are not widely used in practice due to their long degradation time, readily biodegradable high concentrations and rigorous cultivation conditions [24].

In this study, a novel, highly efficient degrading bacterial strain, *Arthrobacter globiformis* DC-1—which could utilize DDT as a sole carbon and energy source—was isolated from long-term DDT-contaminated agricultural soil, and its degradation ability for DDT was examined in pure cultures. The specific objectives were: (1) to characterize the ability of the DC-1 strain to degrade DDT in the MSM; (2) to optimize the parameters of the substrate concentration, temperature, pH, available carbon on the growth and biodegradation of DDT; (3) to examine the DDT congeners degradation ability of the DC-1 strain in pure culture; (4) to explore the DDT intermediate metabolites and potential biodegradation mechanism. To the best of our knowledge, no research has been conducted to identify the intermediate metabolites and the mechanism of DDT degradation using *A. globiformis*.

## 2. Materials and Methods

### 2.1. Chemicals and Culture Media

Standard samples of p,p'-DDT, p,p'-DDE, p,p'-DDD and o,p'-DDT were bought from J&K Chemicals (Beijing, China). All organic solvents and chemicals used in the experiment were of analytical grade.

MSM (4.0 g $NaNO_3$, 1.5 g $KH_2PO_4$, 0.5 g $Na_2HPO_4$, 0.005 g $FeCl_3$, 0.01 g $CaCl_2$ and 0.20 g $MgSO_4$ per liter, pH 7.0) and a MSM agar plate were used for the enrichment

and isolation of DDT-degrading strains. Luria-Bertani (LB) medium (10.0 g NaCl, 10.0 g peptone, 5.0 g yeast extract per liter, pH 7.0) was used for growing the microbial consortium. All of the media were sterilized in an autoclave at 121 °C for 20 min.

### 2.2. Enrichment and Isolation of DDT-Degrading Microorganisms

Soil samples were collected from DDT-contaminated agricultural soil located in the Shenyang North New Area, China. These soil samples were collected in the surface of 0–20 cm using the quincunx sampling method, sifted (2 mm mesh size) to remove the stones and debris and mixed homogeneously to obtain a complex soil sample. The DDT concentration of this soil sample was 137.1 µg/kg. A homogenized soil sample (2 g) was mixed thoroughly with 100 mL of MSM containing 100 mg/LDDT as the sole carbon and energy source. After 4 days of incubation at 30 °C on a rotary shaker at 160 rpm, the culture medium had become turbid. Next, a 1-mL aliquot of soil supernatant was transferred to 100 mL of fresh MSM containing 200 mg/LDDT and was incubated for an additional 4 days under the same conditions. This gradual acclimatization process was repeated three times with increasing concentrations of DDT, ranging between 200 and 500 mg/L, at regular intervals. The final enriched cultures were appropriately diluted and spread on MSM agar plates containing DDT, and then incubated for 4 days at 30 °C. The different colonies that formed were isolated and then tested for their DDT-degrading ability. The bacterial strain that showed the highest DDT degradation ability was named DC-1, and it was used for the subsequent experiments.

### 2.3. Identification of the DC-1 Strain

The new isolated bacterial strain, DC-1, was characterized and identified on the basis of its morphological and biochemical properties according to Bergey's manual of systematic bacteriology, combined with a 16S rDNA gene sequence analysis [25]. The morphological properties of the DC-1 cells were observed through light microscopy (OlymusDM-71, Kyoto, Japan). The 16S rDNA gene sequence of the DC-1 strain was accomplished by the TaKaRa Biotechnology Limited Company, Dalian, China. Similarity analyses were conducted using the NCBI BLAST search. The phylogenetic tree was analyzed using the MEGA software, version 6.06.

### 2.4. Inoculum Preparation

The DC-1 strain was pre-cultured in LB medium at 30 °C and 160 rpm until it reached the exponential phase. DC-1 cells were harvested through centrifugation at 8000 rpm for 5 min, washed three times with sterile distilled water and resuspended in sterile distilled water to obtain an $OD_{600}$ of 1.0 as inoculum.

### 2.5. The Growth and Biodegradation Characteristics of the DC-1 Strain Cultivated in the MSM Supplemented with DDT

In order to study the growth and biodegradation characteristics of the DC-1 strain, an inoculum (10%, *v/v*) was inoculated into MSM containing DDT as the sole carbon and energy source. The samples were incubated at 30 °C on a rotary shaker operating at 160 rpm in the dark. The entire culture was tested for the determination of DDT residue using a gas chromatography-electron coupled with capture detector(GC-ECD; Varian CP 3800, Palo Alto, CA, USA), and the analysis of the bacterial biomass level with an ultraviolet spectrophotometer (UVmini-1240, Kyoto, Japan) at optical density of 600 nm ($OD_{600}$) after inoculation. Each treatment was carried out in triplicate, and the control experiment without the DC-1 strain was carried out under the same conditions.

To assess the growth and DDT-degrading ability of the DC-1 strain, the MSM was supplemented with 10 mg/LDDT, and the incubated samples were collected after 1, 3, 5, 7 and 10 days. To optimize the growth and DDT degradation parameters of the DC-1 strain, the effect of the DDT concentration, temperature, pH and additional carbon sources was tested after 1 day of incubation. The MSM containing DDT at five levels of 1, 5, 10, 20 and

30 mg/L DDT were evaluated to investigate the effect of the initial concentration of DDT on the growth and biodegradation. To check the effect of temperature on the growth and biodegradation, the MSM containing the optimal DDT concentration was incubated at 20, 25, 30, 35 and 40 °C, respectively. Under the optimal DDT concentration and temperature, the MSM was prepared with buffers at pH 4.0, 6.0, 7.0, 7.5, 8.0 and 9.0 for the measurement of the pH effect on the growth and biodegradation. To further investigate the influence of the additional carbon source on the growth and biodegradation under the above optimal conditions, the MSM was supplemented with a 0.5% level of glucose, sucrose, fructose, yeast extract and peptone, respectively.

### 2.6. DDT Congeners Degradation Ability of the DC-1 Strain in Pure Culture

To determine whether the DC-1 strain could degrade additional DDT congeners under the optimal conditions, an inoculum (10%, *v/v*) was inoculated into MSM supplemented with DDD, DDE and o,p'-DDT at final concentrations of 10 mg/L. The samples were incubated on a rotary shaker at 160 rpm in the dark. The whole culture was sampled for the determination of DDT residue after 1, 5 and 10 days of inoculation. Each treatment was performed in triplicate, and the control experiment, which did not include the DC-1 strain, was performed under the same conditions.

### 2.7. Extraction and Analysis of DDT and DDT Congeners

To determine the concentration of DDT and DDT congeners in the MSM, the cultures were extracted three times in equal volumes of n-hexane. The organic phase was then dried on anhydrous sodium sulfate. The sample was then concentrated on a rotary evaporator, and the precipitate was redissolved in a known volume of analytical grade n-hexane for GC analysis.

The quantification of the DDTs was conducted using a GC-ECD and a CP-Sil 8 CB capillary column (30 m × 0.32 mm × 0.25 μm). The operating conditions were described by Wang et al. [18].

### 2.8. Analysis of DDT Metabolites by GC/MS

Experiments identifying and determining DDT metabolites through the DC-1 strain were conducted in MSM supplemented with 10 mg/LDDT at pH 7.5 and a temperature of 30 °C. After 1 day of incubation, the DDT metabolites in the MSM were extracted according to the method described by Pan et al. [26].

The DDT metabolites were analyzed usinga Trace GC Ultra coupled with a Polaris Q ion trap mass spectrometer (Thermo, Waltham, MA, USA). A 1 μL sample was injected and separated on a TR-5MS capillary column (30 m × 0.25 mm × 0.25 μm). Helium acted as the carrier gas at a constant flow rate of 1 mL·min$^{-1}$. The oven temperature program: the oven temperature was initially set to 70 °C for 1 min, then heating at 1 °C·min$^{-1}$ to 76 °C for 0 min, and finally 5 °C·min$^{-1}$ to 330 °C for 10 min. The sample was injected in the split mode with split ratio 50:1 (splitless time: 0.75 min). The temperatures of the injector port, interface and ion source were set at 230, 250 and 250 °C, respectively. The MS detection was in the multiple reactions monitoring (MRM) mode at an electron impact energy of 70 eV. The data were collected using in full-scan (*m/z* 50–550) and scan acquisition with a solvent delay of 3 min.

### 2.9. Data Analysis

The data represent the means and standard deviations (±SD) of three independent experiments. Statistical analysis was conducted using the SPSS 18.0 software, and Graphical Prims 8.3.0 was used to draw the figures.

## 3. Results and Discussion

### 3.1. Identification of the DC-1 Strain

A bacterial DC-1 strain capable of using DDT as the sole carbon and energy source was isolated from long-term DDT-contaminated agricultural soil using an enrichment culture method. The DC-1 strain degraded over 75% of the DDT at a concentration of 10 mg/Lin the MSM in the first day (Figure 1a). The growth response and degradation efficiency of the *A. globiformis* DC-1 in the MSM containing 10 mg/LDDT as the sole carbon and energy source at pH 7.0 and 30 °C were investigated at different time intervals (1, 3, 5, 7 and 10 days). The time course of the growth characteristics and DDT metabolism by *A. globiformis* DC-1 are shown in Figure 1a. During 1 day of incubation with DDT, the DC-1 strain adapted quickly, and its bacterial biomass peaked after only 1 day. The rapid degradation of DDT by the DC-1 strain was observed during 1 day of incubation, reaching up to 76.3%. However, no DDT degradation was observed in the control group that did not receive inoculation. This result demonstrated that the DC-1 strain could grow and metabolize DDT as its sole carbon and energy source in a single day. The DC-1 strain's growth then slowed dramatically after 3 days, and as the incubation time passed, the degradation rate increased very slowly. Due to the abundance of DDT as a carbon source in the MSM, the DC-1 strain degraded the DDT quickly at first. With the continuous degradation, almost all of the DDT was depleted, and the growth of the DC-1 strain lacked a sufficient carbon source, resulting in a sharp decrease in the DDT degradation rate.

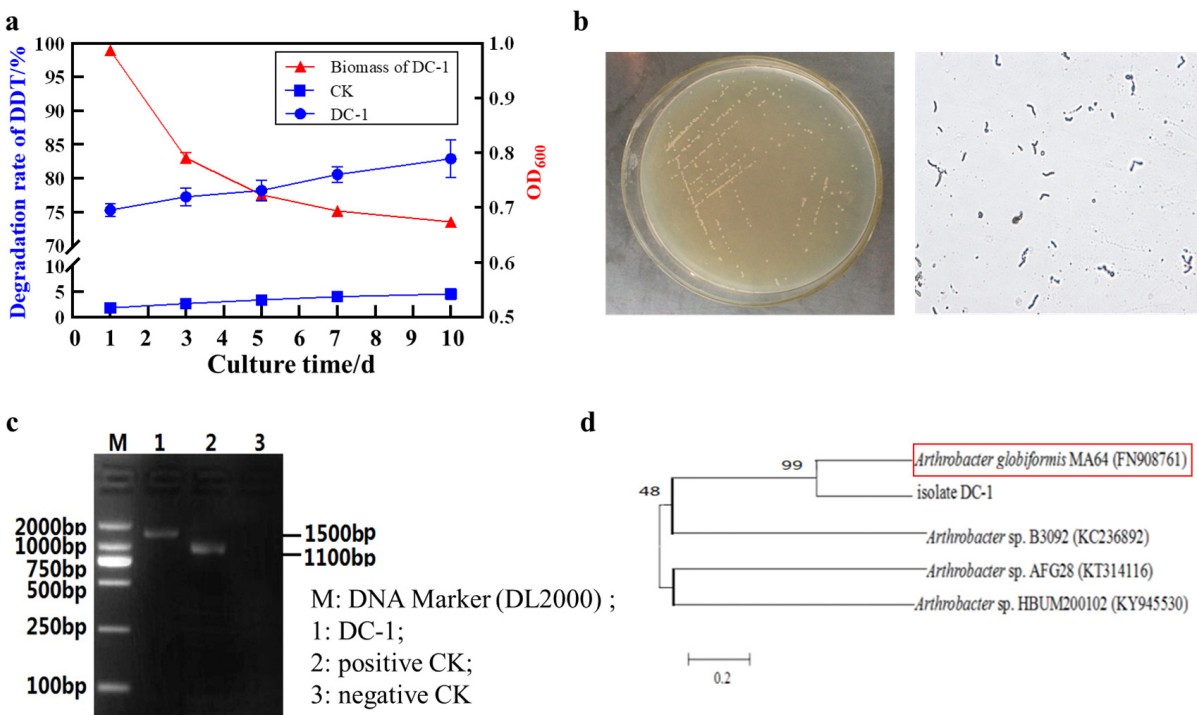

**Figure 1.** (**a**) Growth curve of DC-1 and its DDT degradation rate; (**b**) Colony morphology of DC-1; (**c**) Agarose gel electrophoresis of the DC-1 strain 16S rDNA PCR product; (**d**) Phylogenetic tree based on the 16S r DNA sequences of the DC-1 strain and relating species.

The DC-1 strain was a gram-positive, obligate aerobic and young bacterium, which are short and almost rod shaped (Figure 1b). The colony morphology of DC-1 on the plain agar plate was light yellow, smooth, opaque, wet and with a slightly raised surface (Figure 1b). Biochemical tests showed that the DC-1 strain was positive for oxidase, urease, catalase, nitrate reduction, starch hydrolysis and acetylmethylcarbinol production, but negative for indole production. The DC-1 strain was further studied using the 16S rDNA sequence in the GenBank database (Figure 1c); the DC-1 strain showed the highest sequence similarity

(≥99%) to members of *Arthrobacter globiformis.* Based on the morphological, biochemical and phylogenetic analysis of the bacterial 16S rDNA sequence in GenBank, the DC-1 strain was identified to be *Arthrobacter globiformis* (Figure 1d).

Most microorganisms have been reported to primarily co-metabolize DDT under aerobic conditions [27]: *Trametes versicolor* U97 (cometabolite of 73% in 40 days) [28]; *Methylovorus* sp. XLL03 (possess 50.4% DDT degradation activity with 4 days in the co-presence of glucose) [24]; *Streptomyces* sp. D3 (degrade up to 77% DDT in yeast mineral salt medium after incubation for 7 days) [20]. However, they were unable to use DDT as a sole carbon and energy source. Compared to all of the previous studies, *A. globiformis* DC-1, which was isolated in this study, has the shortest degrading period (1 day) and a higher degradation efficiency (76.3%). Hence, the DC-1 strain is the first *A. globiformis* strain reported to degrade DDT in pure culture and in a shorter time.

### 3.2. The Effect of Environmental Conditions on the Growth and Biodegradation of DDT by the DC-1 Strain

The growth of microorganisms and their biodegradation abilities are largely influenced by various environmental conditions [29,30]. In this study, the impact of the DDT concentration, temperature, pH and carbon source on the *A. globiformis* DC-1 growth and biodegradation rate were investigated.

#### 3.2.1. The Effect of DDT Concentration

The effect of the initial DDT concentrations on the growth response and degradation rate was studied at concentrations ranging between 1–30 mg/L (Figure 2a). The growth and degradation rate of DDT by *A. globiformis* DC-1 increased rapidly as the initial DDT concentration increased, reaching the maximum when the DDT concentration was 10 mg/L. A similar result was observed by Pan et al. [31], in which the degradation rate of DDT by *Ochrobacterium* sp. DDT-2 was increased with the increasing initial concentration (0.1–10 mg/L). In the current study, increasing the DDT concentration to 20 and 30 mg/L inhibited the growth of the DC-1 strain because such high concentrations of DDT were toxic to the DC-1 cells. However, the DDT-induced cells of the DC-1 strain could degrade high concentrations of DDT, reaching 64.0% and 51.2% of the total DDT at concentrations of 20 and 30 mg/L. The results were consistent with Xie et al. [32], who reported that after 3 days of incubation, *Alcaligenes* sp. KK showed increased degradation with the increasing concentrations of DDT, ranging between 1 and 10 mg/L. Similarly, the growth of *Alcaligenes* sp. KK was inhibited by DDT when the concentration exceeded 20 mg/L; Pant et al. [33] and Gao et al. [34] also found that the degrading potential of the strain *Bacillus* GSS and *Alcaligenes* sp. DG-5 for DDT were inhibited at high concentrations (>20 mg/L). In general, DDT-utilizing microorganisms are known to have antimicrobial properties at high concentrations of DDT (>20 mg/L), and only a limited number of microbial strains exhibit a higher degradation capacity of DDT [32]. In this regard, *A. globiformis* DC-1 showed higher tolerance and a degradation capability of up to 30 mg/LDDT as the sole carbon and energy source.

#### 3.2.2. The Effect of Temperature

Temperature has a significant impact on the survival of microorganisms. Low temperatures may inhibit microbial enzyme and metabolism activity, while high temperatures may cause protein inactivation in microorganisms. The different temperatures used in this study showed significant impacts on the growth and biodegradation of DDT by *A. globiformis* DC-1 (Figure 2b). The growth and degradation of DDT by *A. globiformis* DC-1 at temperatures ranging between 25~35 °C were quite promising, with an optimum temperature of 30°C. The ANOVA analysis showed that the growth and degradation rates at 30 °C were significantly higher ($p < 0.05$) than those at 20 °C and 40 °C, indicating that the mesophilic situation was best suitable for the growth and biodegradation of DDT. Similarly, Murtala et al. [9] reported that the growth of the *Staphylococcus* sp. MY 83295F strain was

optimum at a temperature of 30 °C, and the biodegradation of DDT was lower when the temperature was either below or above 30 °C. Mishra and Singh [35] also discovered that 30 °C in broth medium was the best temperature for the maximum degradation rate of DDT by the *Achromobacter* sp. Y12A strain.

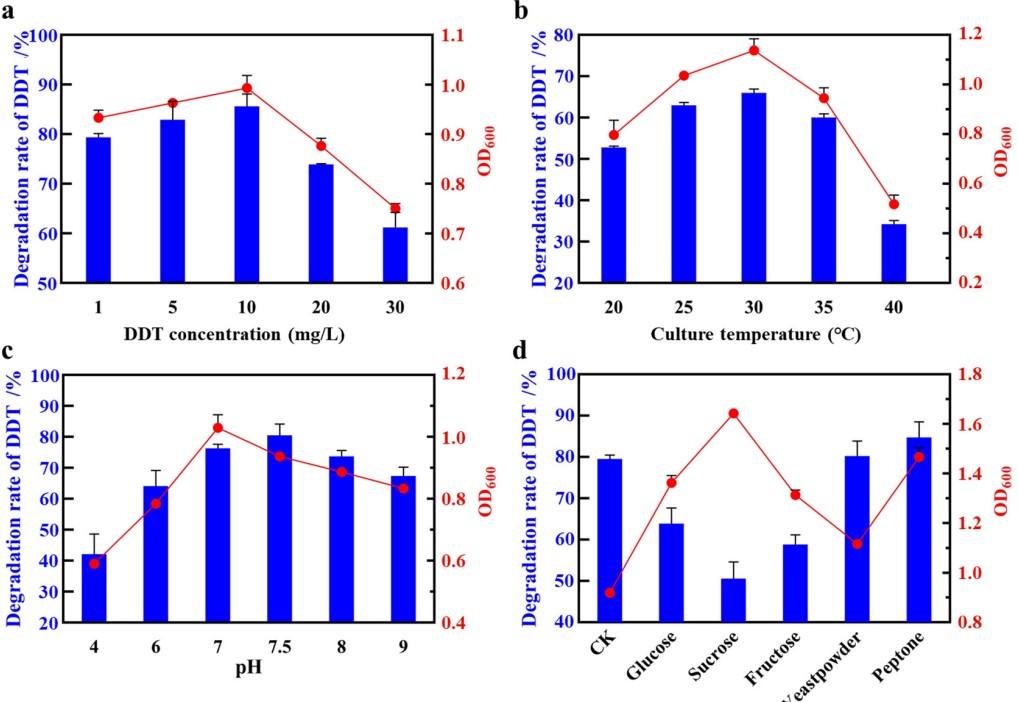

**Figure 2.** Effect of (**a**) DDT concentration, (**b**) culture temperature, (**c**) pH and (**d**) additional carbon sources on DDT degradation and growth of the DC-1 strain.

### 3.2.3. The Effect of pH

To investigate the effect of the pH on the growth and DDT biodegradation, *A. globiformis* DC-1 was inoculated into MSM at a pH range of 4.0–9.0 (Figure 2c). The DC-1 strain had better pH adaptability, and the biodegradation was fairly good throughout the experiment. The rate of the growth and DDT degradation increased with the increasing pH in acidic conditions and decreased in alkaline conditions. The maximum growth was observed at pH 7.0, while maximum the degradation was observed at pH 7.5 (80.51%). The results indicated that neutral (7.0) and alkalescent (7.5) conditions were particularly suitable for DDT biodegradation by *A. globiformis* DC-1, and the DC-1 strain showed more tolerance and DDT degradation at alkaline pH than that of acidic pH. A similar result was disclosed by Powthong et al. [36], who showed that the neutral condition was favorable for the growth of the *Bacillus firmus* S39 strain in minimum medium. Wang et al. [37] also reported that the highest DDT degradation capability of *Pseudoxanthomonas* sp. wax was in the alkalescent condition (pH 7.5).

### 3.2.4. The Effect of Additional Carbon Sources

The effect of supplying various carbon sources on the growth and biodegradation of DDT by *A. globiformis* DC-1 is presented in Figure 2d. The degradation rate of DDT by *A. globiformis* DC-1 was nearly 80.0% after 24 h in the control treatment with no supplementary carbon source. After the addition of glucose, sucrose and fructose, the biodegradation of DDT by *A. globiformis* DC-1 was shown to be inhibited significantly, and only 50.6–63.9% degradation was observed; this could be due to a competitive relationship between the DDT and the co-presence of glucose, sucrose or fructose. Because the DC-1 strain preferred easily metabolizable carbon sources (glucose, sucrose, or fructose) for growth, its preference affected the DDT degradation rate. The degradation effect of adding yeast extract was

similar to that of the control treatment with no additional carbon source, while the co-presence of peptone was considered to be the most suitable carbon source for *A. globiformis* DC-1, and the degradation rate of DDT reached up to 84.2%. A similar phenomenon was observed by Bidlan and Manonmani [38], who reported that the degradation rate of DDT by *Serratia marcescens* DT-1P was enhanced in the co-presence of peptone, but it was inhibited in the co-presence of glucose or sucrose. As shown in Figure 2d, the growth of the DC-1 strain in the presence of any of the five carbon sources was significantly increased than in the control treatment without an additional carbon source ($p < 0.05$). Therefore, the presence of an additional carbon source was not beneficial for the DDT degradation rate by *A. globiformis* DC-1.

### 3.3. DC-1 Strain's Ability to Degrade DDT Congeners in Pure Culture

DDE and DDD are the stable by-products of DDT conversion during biotic and abiotic processes, and they are more persistent and potentially eco-toxic in the environment than the parent compound DDT. These two persistent organic pollutants and o,p'-DDT are commonly cited as sources of impurities in DDT production. Only a few DDT-degrading strains can simultaneously degrade DDE, DDD and DDT as the sole carbon and energy source. Thus, the utilization of the DC-1 strain would be the best option for the degradation of DDT congeners. The biodegradation time course of DDE, DDD and o,p'-DDT by the DC-1 strain is shown in Figure 3. The results indicated that the DC-1 strain had a high utilization capacity for DDT congeners. During 10 days of incubation, the degradation rates of DDE, DDD and o,p'-DDT by the DC-1 strain were up to 70.61%, 64.43% and 60.24%, respectively. To date, only a few studies have described the isolation of DDT-transforming microorganisms from the environment that are available on the removal of DDT and DDT congeners due to their poor bioavailability [34]. The DC-1 strain is the first reported DDT-degrading strain belonging to *Arthrobacter globiformis*, and the first strain capable of degrading DDT and DDT congeners as a sole carbon and energy source.

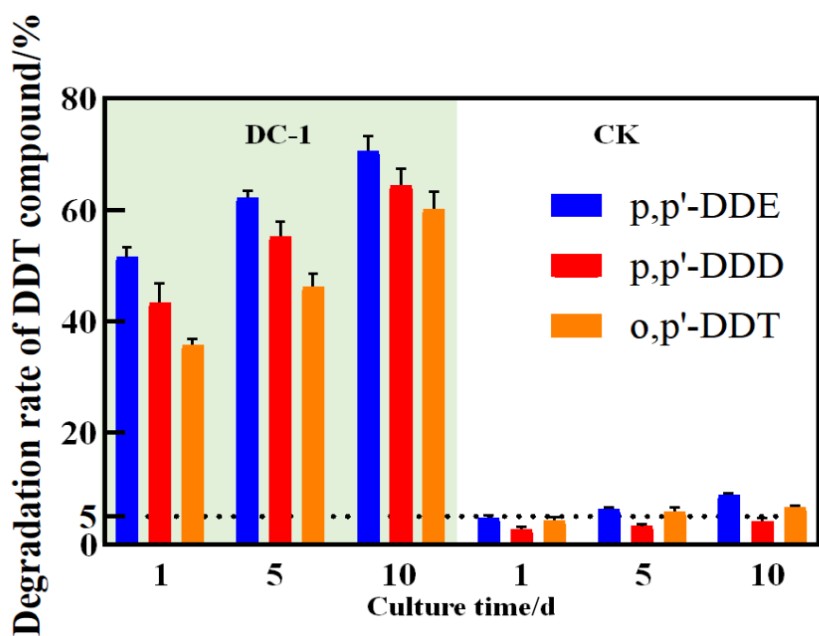

**Figure 3.** The biodegradation of DDT compound by the DC-1 strain.

### 3.4. Identification of DDT Metabolites by GC/MS Analysis

The identification of the metabolites of DDT during its degradation by the bacteria *A. globiformis* DC-1 was carried out in the MSM with 10 mg/L of DDT at pH 7.5 and 30 °C in 1 day of incubation. As shown in Figure 4, three major metabolites—namely, DDD, DDE and DDMU—were observed; their retention times were 37.77, 36.15 and 37.61 min, respectively.

Referring to the known standard compounds and comparisons against the mass spectra in the NIST library, the metabolites were identified as DDD at $m/z$ 319.8 ($C_{14}H_{10}Cl_4$, Figure 5a), DDE at $m/z$ 317.9 ($C_{14}H_8Cl_4$, Figure 5b) and DDMU at $m/z$ 284.0 (1-chloro-2,2-bis(4-chlorophenyl) ethylene, $C_{14}H_9Cl_3$, Figure 5c), respectively. This study found no accumulation of DDD, DDE and DDMU during the GC/MS determination, implying that the biodegradation of DDT by *A. globiformis* DC-1 was a complete mineralization process. DDT's metabolites A (DDD) and B (DDE) were both common major metabolic products, and DDE was previously thought to be a dead-end metabolite [39]. Purnomo et al. [40] reported that the *R. pickettii* strain could convert DDT to DDE as an end-product. Xie et al. [32] also demonstrated that DDT was released in the form of DDE by the *Alcaligenes* KK strain. In our study, the accumulations of DDD and DDE were not found. Similarly, Al-Rashed et al. [41] demonstrated that the *Paracoccus* sp. DDT-21 strain degraded DDD and DDE further; however, their secondary metabolites, DDMU and DDA, were identified. Pan et al. [26] used genome annotation and traditional mass spectrometry methods to examine the DDT degradation products produced by the *Stenotrophomonas* sp. strain, revealing that DDT was first converted to DDD and DDE, then to DDMU and DDOH, and finally to DDA. DDA and DDOH were not detected through the GC-MS in this study.

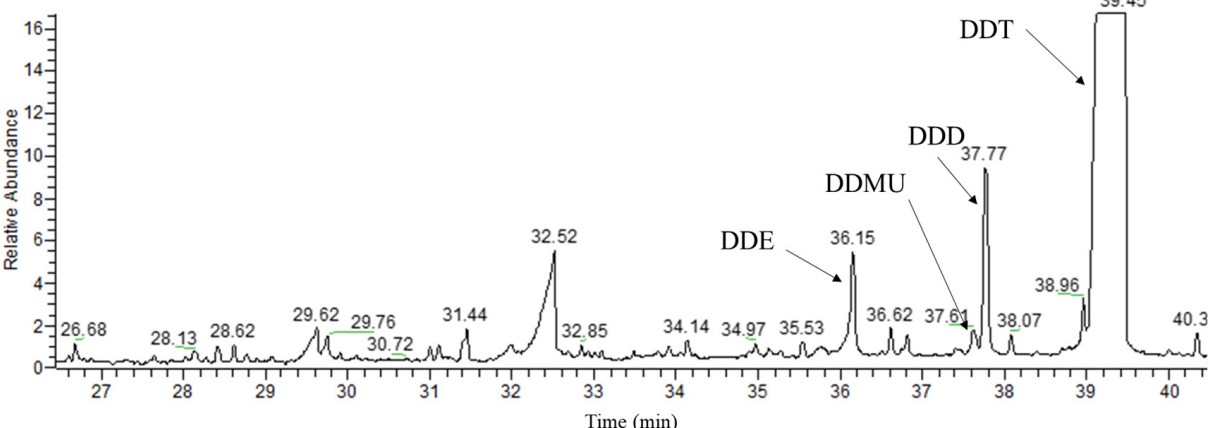

**Figure 4.** Total ion chromatogram (TIC) of DDT degradation products by the DC-1 strain.

A proposed DDT biodegradation pathway based on our identification of the DDT metabolites is presented in Figure 6. The biodegradation of DDT by *A. globiformis* DC-1 consisted of initial dechlorination and dehydrochlorination at the trichloromethyl group into DDD and DDE, respectively. Subsequently, the dechlorination reaction took place again to produce DDMU. Finally, after the ring opening reaction, the DDMU was mineralized to carbon dioxide step by step. Several similar studies on the biodegradation pathway of DDT were carried out using GC-MS. Sariwati and Purnomo [42] reported that mixed cultures of *Fomitopsispinicola* and *Pseudomonas aeruginosa* initially transformed DDT to DDE and DDD, and then to DDMU, as metabolic products. Feng et al. [24] also reported a mineralization process of DDT by the *Methylovorus* sp. XLL03 strain. DDT could first be dechlorinated to form DDD and DDE, subsequently dechlorinated to DDMU, and then completely mineralized though meta-ring cleavage reactions. Similarly, Bajaj et al. [43] reported DDT metabolism into DDE, DDD and DDMU in culture broth by the *Rhococcus* sp. IITR03 strain.

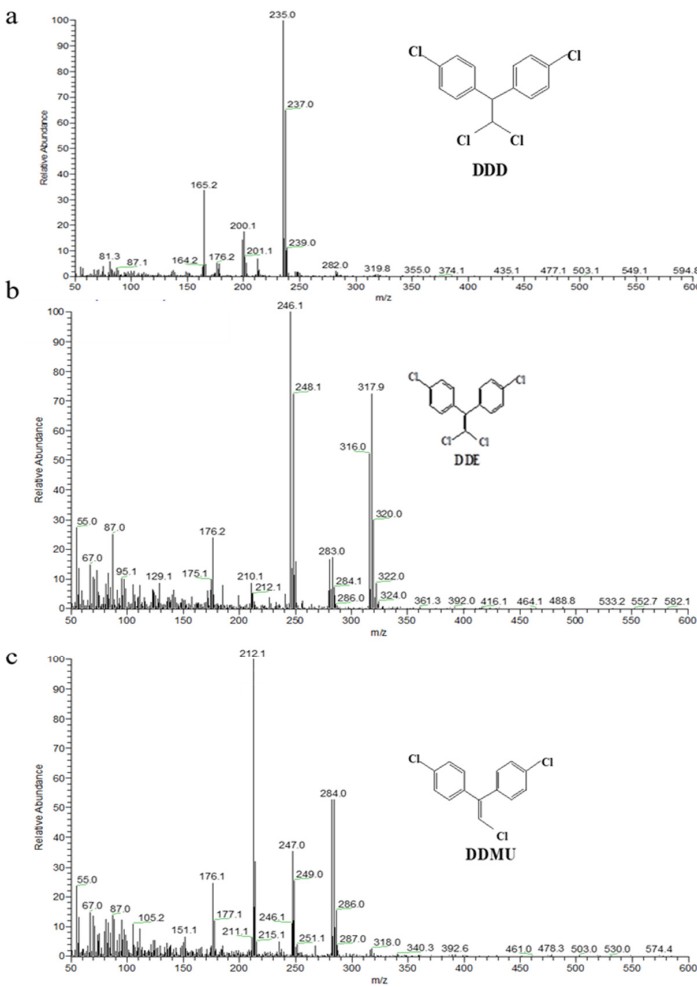

**Figure 5.** Mass spectrogram and proposed structures of DDT degradation products (**a**) (DDD), (**b**) (DDE) and (**c**) (DDMU).

**Figure 6.** Metabolic pathway of DDT degradation by the DC-1 strain.

## 4. Conclusions

An enrichment culture method was used to isolate a bacterial DC-1 strain capable of degrading DDT as the sole carbon and energy source from DDT-contaminated agricultural soil, and it was identified as belonging to *Arthrobacter globiformis*. The DC-1 strain could degrade 76.3% of the DDT at a concentration of 10 mg/L in MSM in the shortest degrading period (1 day). The growth and degradation rate of DDT by *A. globiformis* DC-1 increased with the increasing concentration, ranging between 1 and 10 mg/L, but decreased with the increasing concentration, ranging between 20 and 30 mg/L. The *A. globiformis* DC-1 strain could efficiently grow and degrade at temperatures ranging between 20 and 40 °C, and at pH ranging between 4.0–9.0. The maximum DDT degradation (~84.2%) was observed at 10 mg/LDDT, with peptone as the carbon source and pH 7.5 at 30 °C for 1 day of incubation. The DC-1 strain could also effectively degrade DDE, DDD and o,p'-DDT as the sole carbon and energy source. The strain is the first *Arthrobacter globiformis* DDT-degrading strain reported that is capable of degrading DDT and DDT congeners as a sole carbon and energy source. During the degradation process, DDD, DDE and DDMU were detected as metabolic products. Based on the identification of these metabolites, a DDT biodegradation pathway was proposed. This study showed that the *A. globiformis* DC-1 strain can be a viable option for the bioremediation of DDT residues in the environment. Further studies on the behavior of microorganisms and the effect of environmental conditions need to be considered carefully, and some strengthening measures (surfactant, plant etc.) should be utilized in an effort to explore more effective bioremediation.

**Author Contributions:** X.W.: Writing—original draft; B.T.O.: Writing—review and editing; H.W. (Hui Wang): Writing—original draft; Q.L.: Funding acqusion, supervision; J.L.: Figures; L.T.: Formal analysis; M.Y.: Formal analysis ; H.W. (Hao Wu): Writing—review and editing; L.S.: Writing—review and editing. All authors have read and agreed to the published version of the manuscript.

**Funding:** This research was funded by the Applied Basic Research Program of Liaoning Province (NO. 2023JH2/101300015), the Young and Middle-aged Scientific and Technological Innovation Talents Project of Shenyang (NO. RC220128), the funding project of Northeast Geological S&T Innovation Center of China Geological Survey (NO. QCJJ2022-29), and the China Postdoctoral Science Foundation (NO. 2022MD72379).

**Data Availability Statement:** All the research data has been provided in the paper.

**Acknowledgments:** The authors would like to thank laboratory technicians for their help.

**Conflicts of Interest:** The authors declare that they have no known competing financial interest or personal relationship that could have influenced the work reported in this paper.

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
