# Peer review of "Degradation of DDT by a Novel Bacterium, Arthrobacter globiformis DC-1: Efficacy, Mechanism and Comparative Advantage"

_water, doi:10.3390/w15152723_

Round 1

Reviewer 1 Report

Abstract - line 24, no need for a multiplication mark in mgL-1

Introduction gives sufficient amount of the information about the current findings regarding DDT polution and biodegradation. Article objectives are clearly stated.

Used materials and methods were described in detail, and make it possible for other researchers to repeat the process if needed.

All environmental variables were taken into account and results are clearly presented.

Conclusion nicely summarizes all the findings.

I have no remarks regarding this article. I think article will enhance overall knowledge in the field of bioremediation and is worth publishing.

English is ok, please make sure to make grammar and spell check once more.

Author Response

Dear Reviewer:

Thank you for your comments concerning our manuscript entitled “Degradation of DDT by a novel bacterium, Arthrobacter globiformis DC-1: Efficacy, mechanism, and comparative advantage” (ID: water-2527999). The comments are all valuable and very helpful for revising and improving our paper, as well as the important guiding significance to our researches. We have studied comments carefully and have made correction which we hope meet with the approval. Revised portion are marked in red in the paper. The main corrections in the paper and the respond to the reviewer’s comments are as followings:

Respond to the reviewer’s comments:

First of all, thank you for your affirmation and suggestion.

Abstract - line 24, no need for a multiplication mark in mg·L-1

Response: We are very sorry for our incorrect writing.We have revised them, and mg·L-1 had been replaced with mg/L according to the reviewer’s suggestion marked in red in the manuscript.

Once again, thank you very much for your comments and suggestion.

Thank you and best regards.

Yours sincerely,

Xiaoxu Wang

Name: Hui Wang and Qing Luo

Reviewer 2 Report

The work concerns an isolated bacterial strain capable of degrading DDT as the only source of carbon and energy from agricultural soil contaminated with DDT. The work is well-planned and described. The results regarding the degradation of DDT and its derivatives are promising. The work will be interesting for readers. The reviewer offers suggestions to improve the manuscript:

·        Line 81 replace „condition” with conditions.

·        2.2. Enrichment and isolation of DDT-degrading microorganisms - What mass of soil was taken for testing? How was representative soil obtained? The authors write that the soil was contaminated with DDT, what was the concentration in the soil? Please complete the information in the manuscript.

·        Line 148 replace „1 day incubation” with 1-day of incubation.

·        Line 263 replace „has showed” with has showed.

·        Line 292 replace „alkalescency condition” with alkalescent condition.

·        Line 296 replace „were presented: with was presented.

·        Line 310 replace „were” with was.

·        Figure 5 –please improve the figure quality.

·        Conclusion - what are the further research plans for finding effective ways to degrade DDT?

Minor editing of English language required.

Author Response

Dear Reviewer:

Thank you for your comments concerning our manuscript entitled “Degradation of DDT by a novel bacterium, Arthrobacter globiformis DC-1: Efficacy, mechanism, and comparative advantage” (ID: water-2527999). The comments are all valuable and very helpful for revising and improving our paper, as well as the important guiding significance to our researches. We have studied comments carefully and have made correction which we hope meet with the approval. Revised portion are marked in red in the paper. The main corrections in the paper and the responds to the reviewer’s comments are as followings:

Responds to the reviewer’s comments:

  1. Line 81 replace “condition” with “conditions”.

Response: We are very sorry for our incorrect writing. We have revised it, and “condition” have been replaced with “conditions” according to the reviewer’s suggestion marked in red in the manuscript.

  1. 2.2. Enrichment and isolation of DDT-degrading microorganisms -What mass of soil was taken for testing? How was representative soil obtained? The authors write that the soil was contaminated with DDT, what was the concentration in the soil? Please complete the information in the manuscript.

Response: We are very sorry for our negligence, now the details in the soil sample have been added.

After modification: Soil samples were collected from a DDT contaminated agricultural soil located in the Shenyang North New Area, China. These soil samples were collected in the surface of 0-20 cm by quincunx sampling method and sifted (2mm mesh size) to remove the stones and debris and mixed homogeneously to get a complex soil sample. The DDT concentration of this soil sample was 137.1 μg/kg. A homogenized soil sample (2 g) was mixed thoroughly with 100 mL of MSM containing 100 mg/L DDT as the sole carbon and energy source.

  1. Line 148 replace “1 day incubation” with “1-day of incubation”.

Response: We are very sorry for our incorrect writing.We have revised them, and“ 1 day incubation” have been replaced with“ 1-day of incubation” according to the reviewer’s suggestion marked in red in the manuscript.

  1. Line 263 replace “has showed” with “had showed”.

Response: We are very sorry for our incorrect writing. We have revised it, and “has showed ”have been replaced with “had showed” according to the reviewer’s suggestion marked in red in the manuscript.

  1.  Line 292 replace “alkalescency condition” with “alkalescent condition”.

Response: We are very sorry for our incorrect writing. We have revised them, and“ alkalescency condition” had been replaced with “alkalescent condition” according to the reviewer’s suggestion marked in red in the manuscript.

  1. Line 296 replace “were presented” with “was presented”.

Response: We are very sorry for our incorrect writing. We have revised it, and“were presented” had been replaced with “was presented” according to the reviewer’s suggestion marked in red in the manuscript.

  1. Line 310 replace “were” with “was”.

Response: We are very sorry for our incorrect writing. We have revised it, and“were” had been replaced with “was”according to the reviewer’s suggestion marked in red in the manuscript.

  1.  Figure 5 –please improve the figure quality.

Response: Thank you for your comment. We have improve the figure 5 quality.

  1. Conclusion - what are the further research plans for finding effective ways to degrade DDT?

Response: We are very sorry for our negligence. We have added the further research plans in conclusion.

After modification: Further studies on the behavior of microorganism and the effect of environmental conditions need to be considered carefully, and some strengthening measures (surfactant, plant etc.) should be utilized in an effort to explore more effective bioremediation.

Once again, thank you very much for your comments and suggestions.

Thank you and best regards.

Yours sincerely,

Xiaoxu Wang

Name: Hui Wang and Qing Luo
